# The Role of Intraventricular Hemorrhage in Traumatic Brain Injury: A Novel Scoring System

**DOI:** 10.3390/jcm11082127

**Published:** 2022-04-11

**Authors:** Cheng-Yu Li, Chi-Cheng Chuang, Ching-Chang Chen, Po-Hsun Tu, Yu-Chi Wang, Mun-Chun Yeap, Chun-Ting Chen, Ting-Wei Chang, Zhuo-Hao Liu

**Affiliations:** Department of Neurosurgery, Chang Gung Memorial Hospital, Linkou Medical Center, Chang Gung University, Taoyuan 333, Taiwan; mick791212@hotmail.com (C.-Y.L.); ccc2915@cgmh.org.tw (C.-C.C.); 8702047@cgmh.org.tw (C.-C.C.); d12096@cgmh.org.tw (P.-H.T.); m7849@cgmh.org.tw (Y.-C.W.); m0125@cgmh.org.tw (M.-C.Y.); b9002055@cgmh.org.tw (C.-T.C.); b9202045@cgmh.org.tw (T.-W.C.)

**Keywords:** intraventricular hemorrhage, traumatic brain injury, Traumatic Graeb Score

## Abstract

Traumatic intraventricular hemorrhage (tIVH) is associated with increased mortality and disability in traumatic brain injury (TBI). However, the significance of tIVH itself remains unclear. Our goal is to assess whether tIVH affects in-hospital mortality and short-term functional outcomes. We retrospectively reviewed the records of 5048 patients with TBI during a 5-year period, and 149 tIVH patients were analyzed. Confounding was reduced using the inverse probability of treatment weighting (IPTW) based on propensity score. The association between IVH and outcomes was investigated using logistic regression in the IPTW-adjusted cohort. In our study, after adjustment for analysis, the in-hospital mortality rate (11.4% vs. 9.2%) and the poor functional outcome rate (37.9% vs.10.6%) were significantly higher in the tIVH group than in the non-tIVH group. Factors independently associated with outcomes were age ≥ 65 years, Glasgow Coma Scale (GCS) severity score, and the Graeb score. The Traumatic Graeb Score, a novel scoring system for predicting functional outcomes associated with tIVH, comprised the sum of the following components: GCS scores of 3 to 4 (=2 points), 5 to 12 (=1 point), 13 to 15 (=0 points); age ≥ 65 years, yes (=1 point), no (=0 points); Graeb score (0–12 points). A Traumatic Graeb Score ≥ 4 is an optimal cutoff value for poor short-term functional outcomes.

## 1. Introduction

The prevalence of traumatic intraventricular hemorrhage (tIVH) in patients with traumatic brain injury (TBI) ranges from 0.4% to 22%, with higher rates of occurrence associated with more severe brain injury [1,2,3,4,5]. However, the significance of tIVH independent of such brain injury remains unclear. Studies have suggested that tIVH is associated with poor prognosis because of the injury of neurons caused by angular acceleration or shearing force [6]. Other studies have suggested the effect is caused by consequences of the associated injury, because patients of isolated tIVH have also been reported to have highly favorable outcomes [7,8].

The severity of tIVH is reportedly associated with diffuse axonal injury (DAI) lesions in the corpus callosum [9]. However, studies examining the severity of tIVH on initial computed tomography (CT) scans and poor outcomes are scarce, probably because of small available sample sizes due to the relative rarity of the condition [10]. Although the Graeb score, intraventricular hemorrhage (IVH) score, and the LeRoux score have all been validated in IVH related to spontaneous intracerebral hemorrhage (ICH), no grading scale for tIVH is consistently used for the prediction of clinical outcomes [11].

In the current study, patients with TBI presenting with tIVH were examined. In-hospital mortality and short-term functional outcomes in traumatic intraventricular hemorrhage were analyzed. The aim of our study was to create a practical and accurate prognostic model to predict patient outcomes in tIVH and to rapidly assess tIVH at initial presentation.

## 2. Materials and Methods

### 2.1. Study Design and Sample Size Calculation

This was a retrospective cross-sectional study. The calculation of receiver operating characteristic curve analysis requires a minimum of sample size more than 100 in both the case (tIVH) and control (non-tIVH) groups [12]. Given the annual number of patients with blunt head trauma of 1000 and the estimated tIVH rate of 2% in our hospital, we planned to collect 5-year data between 2015 and 2019.

### 2.2. Inclusion and Exclusion Criteria

Medical records of patients who were admitted to the neurosurgical department at a tertiary trauma center between January 2015 and December 2019 were retrospectively reviewed. Patients with missing or incomplete demographic information, patients who died before admission, and patients with incidentally discovered intracranial aneurysms or spontaneous ICH were excluded (Figure 1). The study was approved by our institutional review board (IRB: 20170057B0).

### 2.3. Demographic and Clinical Variables

We analyzed the following demographic and clinical variables: sex, age, presence of comorbidities, Glasgow Coma Scale (GCS) score, systolic blood pressure at emergency department (ED) presentation, injury severity score (ISS), and New Injury Severity Scale (NISS) score. Comorbidities were defined using any inpatient diagnosis before the index date or the presence of any diagnosis on medical records. Data collection was conducted in a standardized manner through the TBI database at a tertiary trauma center.

### 2.4. Radiographic Variables

Admission CT scans were performed on initial visit to the ED. The CT findings included skull fracture (linear or depressed), traumatic subarachnoid hemorrhage (SAH), acute epidural hematoma, acute subdural hematoma, traumatic ICH (cerebral contusion, excluding the typical petechial radiological pattern of DAI), and tIVH. The severity of tIVH was evaluated using three IVH scoring systems—the IVH, the Graeb score, and the LeRoux score—consistent with the approach of related studies [1,10,11,13]. Two experienced neuroradiologists who had been working in a neuroradiology department at a tertiary trauma center for more than 10 years and were blinded to the study outcomes analyzed all CT scans.

### 2.5. Outcome Assessment

The primary outcomes were in-hospital mortality and functional outcomes at discharge using the modified Rankin Scale (mRS). We defined poor functional outcomes as mRS scores > 2. The secondary outcomes were the numbers and types of operations performed (e.g., external ventricular drainage, craniotomy, craniectomy), complications experienced during hospitalization, and lengths of stay in hospital and in the intensive care unit (ICU).

### 2.6. Statistical Analysis

We created an adjusted cohort using inverse probability of treatment weighting (IPTW) with stabilized weight based on propensity scores to reduce confounding variables when comparing outcomes between the tIVH and non-tIVH groups. Compared with propensity score matching, IPTW provides higher statistical power without loss of sample size [14]. The balance of the covariates before and after IPTW between groups was assessed using the standardized difference test (STD), where an absolute value less than 0.2 was considered a small difference [15].

In-hospital outcomes and complications between groups were compared using logistic regression analysis for binary outcomes (i.e., in-hospital mortality) and a linear regression model for continuous outcomes (i.e., admission days). We also performed receiver operating characteristic (ROC) curve analyses to assess the performance of the known scores (the IVH, Graeb score, and LeRoux score) in discriminating the two primary outcomes.

Moreover, patients with tIVH who died or recorded poor outcomes at discharge or not were compared using the univariate analysis (Chi-square test for nominal data; Student’s *t*-test for numerical data). On the basis of the well-known ICH scores and previous TBI studies, we recategorized patients’ demographic data according to age and GCS score at initial presentation at the ED [16,17]. Variables with *p*-value < 0.2 were put into the multivariate logistic regression (MLR) [18]. The performance of the MLR analysis was assessed by determining calibration and discrimination. Calibration was assessed using the Hosmer–Lemeshow Ĉ-test (with *p* > 0.05 indicating no significant difference between the predicted and observed outcomes) [19]. Subsequently, we developed an outcome risk stratification scale (the Traumatic Graeb Score) composed of variables associated with in-hospital mortality and poor functional outcomes, with weighting based on the strength of independent associations. Finally, the discrimination performance between the new scoring systems and the existing IVH grading systems was compared using DeLong’s test.

A two-sided *p* value was considered statistically significant. All statistical analyses were performed using SAS version 9.4 (SAS Institute, Cary, NC, USA) and MedCalc version 20.008 (MedCalc Software, Ostend, Belgium) code.

## 3. Results

### 3.1. Baseline Characteristics of the Study Population

During the study period, 5048 patients with blunt head trauma presented to our ED where a CT scan was ordered. After exclusionary criteria were applied, 5000 patients were enrolled in our study. Of the 5000 patients, tIVH was found in 149 patients, with a prevalence rate of 2.98%. The baseline demographics, clinical variables, and other abnormal head CT findings of the enrolled patients are listed in Table 1.

For the tIVH group, the mean age was 54.2 (standard deviation [SD], 26.0) years; and the mean age was 49.4 (SD, 24.4) years in the non-tIVH group. Lower means of the GCS score (9.1 ± 4.4 vs. 12.5 ± 3.9) and a higher proportion of severe TBI (53.0% vs. 19.7%) were found in tIVH group, which suggest greater clinical severity. Substantial differences in the characteristics before weighting, included GCS score at ED, ISS, NISS, SAH, and ICH (absolute STD value > 0.2). The distribution between the two groups was more balanced after weighting with all of the absolute STD values < 0.2.

### 3.2. Traumatic IVH and Outcomes

In total, 38 (25.5%) tIVH patients and 427 (8.8%) non-tIVH patients died during hospitalization. After IPTW, the presence of tIVH was significantly associated with a higher risk of mortality (11.4% vs. 9.2%; odds ratio [OR], 1.27; 95% confidence interval [CI], 1.11–1.45). Before IPTW, 85 (57%) and 491 (10.1%) patients recorded poor outcomes (mRS > 2) at discharge in the tIVH and non-tIVH groups, respectively. After IPTW, the disability rate at discharge was significantly higher in the tIVH group (37.9% vs. 10.6%; OR, 5.13; 95% CI, 4.60–5.71). In addition, external ventricular drainage insertion was performed much more frequently and craniotomy was performed less frequently in the tIVH group compared with the non-tIVH group. With the exception of gastrointestinal bleeding, patients with tIVH experienced more in-hospital complications, including new onset seizure, arrhythmia, pneumonia, urinary tract infection, and sepsis. These patients also required a longer length of stay in the intensive care unit and hospital (Table 2).

Twelve patients with isolated tIVH were identified in our study. Outcomes were also poor among these patients: seven patients had poor functional outcomes at discharge and three died.

### 3.3. IVH Grading Scales and Outcomes

Appendix A lists the AUCs for the Graeb, LeRoux, and IVH scores related to in-hospital mortality and poor functional outcomes at discharge. All three scores exhibited satisfactory performance in discriminating mortality, with the AUC values as follows: Graeb, 0.847 (95% CI, 0.779–0.901); LeRoux, 0.844 (95% CI, 0.775–0.898); and IVH, 0.828 (95% CI, 0.757–0.885). No significant differences were observed in the AUCs of the three scores. The performance of the three scores in discriminating poor functional outcomes was also favorable, with the AUC values as follows: Graeb, 0.821 (95% CI, 0.750–0.879); LeRoux, 0.793 (95% CI, 0.718–0.854); and IVH, 0.813 (95% CI, 0.741–0.872). The AUC value of the Graeb score was the highest among the three scoring systems and was significantly more effective than the LeRoux score in predicting poor functional outcomes.

### 3.4. Risk Factors for In-Hospital Mortality and Poor Functional Outcome in tIVH

After we recategorized patients according to age older than 65 and GCS score severity at ED (grade 0, GCS score 13–15; grade 1, GCS score 5–12; and grade 2, GCS score 3–4), both factors were significantly associated with mortality and poor functional outcomes; we introduced those variables with a significance level <0.2 in the univariable logistic regression analysis (Table 3) and the Graeb score into the MLR. (Table 4). We selected the Graeb score because its area under the ROC curve (AUC-ROC) was the highest among the three known scores. The independent risk factors for in-hospital mortality in the tIVH group were identified as age ≥ 65, ED GCS severity score, and the Graeb score. The three risk factors were also observed in the results for patients with poor functional outcomes at discharge. Furthermore, this MLR model showed good calibration (Hosmer–Lemeshow Ĉ-test, *p* = 0.816 for in-hospital mortality; *p* = 0.562 for poor functional outcomes).

### 3.5. Novel Traumatic Graeb Score

On the basis of the independent predictors determined from the logistic regression model, we developed an outcome risk stratification scale, the Traumatic Graeb Score. Table 5 lists the three components that were assigned points according to the strength of association with mortality and poor functional outcomes. IVH grading had the largest weighting in the scale. Age ≥ 65 and ED GCS severity score were weighted equally because of similar outcome associations. One point was given for patients older than 65 years. Points for ED GCS severity score ranged from 0 to 2 depending on ED GCS severity grading.

We also applied a similar method, adding age and ED GCS severity score as variables, to develop a Traumatic LeRoux Score and a Traumatic IVH Score. Comparison of the AUC-ROC between these new scores and the previous IVH grading scales are presented in Appendix A and Appendix A. Of the new models, the AUC-ROC of the Traumatic Graeb Score was the only one that was significantly higher than all previous IVH grading scales, both for mortality (AUC = 0.888; 95% CI, 0.826–0.934; Figure 2A) and prediction of poor outcomes at discharge (AUC = 0.880; 95% CI, 0.817–0.928; Figure 2B). The Traumatic Graeb Scores were then examined for accuracy based on the presence of IVH in different ventricles (Appendix A). The AUC values were all above 0.85, suggesting that the Traumatic Graeb Score was applicable to all patients with tIVH.

The total Traumatic Graeb Score is the sum of points (range: 1–15) for various characteristics. No patients in our cohort study obtained the maximum 15 points. Traumatic Graeb scores of ≥4 were associated with maximum Youden Index values, and thus were identified as an optimal cutoff value for the prediction of poor functional outcomes. By contrast, the mortality rates increased from 5.2% in patients with Traumatic Graeb scores of ≤3, to 39.1% for those with scores of ≥4 (Figure 3). No patient with a Traumatic Graeb Score of 1 or 2 died, whereas all patients with a Traumatic Graeb Score higher than 10 died.

## 4. Discussion

The incidence of tIVH in our cohort was consistent with rates in other studies [2,3,20,21,22]. The demographics of the tIVH group were similar to all patients with blunt TBI, with men and middle-aged patients predominating, suggesting that tIVH is not primarily associated with certain patient groups.

The relationship between the presence of tIVH and clinical outcomes has been extensively discussed in the literature. Although most studies have stated that tIVH is associated with poor prognosis, with only 12% to 47% of patients achieving functional recovery [4,23], other studies have indicated no such association after matching for associated intracranial injuries [5,24]. However, growing evidence supports the conclusion that tIVH is associated with poor prognosis, as do the results of our study. The proposed mechanisms for greater disability in patients with tIVH include indirect injuries (shearing strain on midline structures), increased posttraumatic hydrocephalus, and the coexistence of voluminous hemorrhagic mass [8,25,26,27]. Matsukawa et al. first reported that tIVH was significantly associated with DAI lesions located in the corpus callosum [6]. Mata-Mbemba at al. discovered a positive correlation between IVH scores and the severity of DAI [9]. These studies indicated that the outcomes of patients with tIVH could be substantially influenced by the presence of DAI, resulting in higher rates of disability. The other suggested possibility, posttraumatic hydrocephalus, has been reported to be significantly associated with tIVH as a result of obstructed CSF circulation by blood clots or decreased absorption at the arachnoid granulation [25,28]. Our study also revealed a positive correlation, with a higher number of CSF diversion procedures performed in the tIVH group. Regarding the effect of the coexistence of voluminous hemorrhagic mass, some researchers have stated that isolated patients with tIVH had more favorable outcomes and lower mortality than non-isolated patients with tIVH [7,29]. Thus, whether tIVH affects the clinical course directly or is the result of associated brain injury remains unclear. Because different brain lesions, extracranial injuries, and underlying comorbidities often coexist in patients with TBI, we attempted to minimize these confounding effects by performing IPTW during the statistical analysis. In our cohort, the number of patients who underwent craniotomies was significantly lower in the tIVH group than in the non-tIVH group, leading to the speculation that these factors may contribute less to clinical outcomes. By contrast, regarding isolated tIVH, 7 of 12 patients with isolated tIVH in our study also exhibited poor functional outcomes, with 6 of them exhibiting GCS scores of ≤8 at initial presentation in the ED, suggesting that tIVH is independently associated with substantial morbidity.

Although the possible pathophysiology for higher disability and mortality in tIVH patients is complex and related to the context of the severity and evolution of TBI, establishing an effective prognostic model for developing appropriate clinical treatments remains critical. The current IVH grading systems (the Graeb, LeRoux, and IVH scores) were initially designed for spontaneous supratentorial ICH and were then introduced in different clinical scenarios. The LeRoux score was applied to patients with TBI with IVH, whereas the modified Graeb score was developed to assess IVH in patients with aneurysmal SAH with ICH [1,30]. However, studies regarding the contribution of the severity of IVH (in addition to patient characteristics) to the clinical outcomes in patients with TBI are rare. No widely used clinical grading scales for tIVH exist.

In our cohort, we introduced all three widely accepted grading systems to assess the severity of IVH through brain CT scans. Comparison of the power of association with prognosis revealed that the Graeb score was the optimal existing grading system, being capable of balancing simplicity and the accuracy of functional outcome prediction. We therefore proposed a modified and widely applicable clinical grading scale, the Traumatic Graeb Score, after incorporating two other independent clinical prognostic factors. The Traumatic Graeb Score is a straightforward scoring system composed of a neurologic examination (GCS), baseline characteristic (age), and the severity of IVH (the Graeb score). It exhibits favorable predictability for functional outcomes at discharge. The purpose of this grading scale is to provide an assessment tool that health-care providers can use to rapidly assess patients at the time of tIVH presentation, regardless of the coexistence of voluminous hemorrhagic mass, and to allow appropriate treatment selection for neurosurgeons.

The Traumatic Graeb Score is a novel grading scale for the prediction of in-hospital mortality and functional outcomes in patients with tIVH. The findings of our study suggest that with the use of new statistical methods, the severity of tIVH itself should be regarded as an independent predictor of outcome and mortality in patients with TBI. However, the GCS classification and cutoff value for age in the Traumatic Graeb Score are unique to our model and deserve discussion. The GCS score has been a standard and reliable assessment tool for all types of neurologic events. Although most studies examining severity of TBI have classified GCS scores of 3–8, 9–12 and 13–15 into severe, moderate, and mild, an increasing number of studies have discovered that the previous classifications are too rigid and that patients with GCS scores of 3 and 4 had far poorer prognosis than did those with other scores [3,17]. Patients with GCS score ≥ 13, by contrast, exhibited much more favorable functional outcomes. These findings are consistent with the results of our study. Because of the strength of the Traumatic Graeb score’s outcome predictability, reclassifying GCS scores into three groups and weighting the components similarly is justified in the Traumatic Graeb Score.

Increased age has long been recognized as a poor prognostic factor in TBI [31]. However, whether age is associated with in-hospital mortality and neurologic outcomes in tIVH is inconclusive according to the findings reported in the literature [3,7]. Salotto et al. reported that elderly patients with TBI had higher GCS scores than younger patients with TBI with similar levels of severity of TBI [23]. In our study, we found that in patients with tIVH, age ≥ 65 years was independently associated with poorer outcomes. Two possible explanations may account for this finding. First, elderly patients may have more severe neurologic outcomes following brain insult irrespective of the severity of IVH, leading to higher levels of disability. Moreover, if patients have multiple chronic diseases, family members may prefer conservative treatment and refuse surgical intervention even if the IVH-related injury is not as profound. Therefore, studies focusing on aggressive surgical intervention in TBI often exclude older populations. In our cohort, 63 of 149 patients were older than 65 years. Only four patients were provided hospice care because of comorbidities (two were diagnosed as having dementia, and the others were of bed-ridden status), given that they could possibly survive a brain injury if they received suitable neurosurgical intervention. Thus, we believe that the impact of age on risk stratification after tIVH is related to neurologic injury rather than less aggressive medical care decisions.

To our knowledge, our study enrolled the largest cohort for examining tIVH since Atzema’s study in 2006 [7]. We found out that patients with tIVH usually have greater clinical severity and that tIVH presented initially would be a key contributor to patient outcomes at discharge. We considered all semiquantitative tools measuring the severity of IVH and found that increased IVH extension did correlate with poor functional outcomes. A novel risk stratification scale, the Traumatic Graeb Score, composed of GCS severity score, age, and Graeb score at ED, enhances consistency in clinical care and clearer decision making for factors related to tIVH.

Our study has some limitations. First, the sample size of patients was relatively small because it was a single-institution cohort. However, given the rarity of tIVH, our sample size was larger than those in most previously reported studies. Second, patients who died before admission were excluded in our study. Since they were possibly the most severe cases, study results could be biased. Third, although 3- and 6-month functional outcomes using mRS were preferred in other studies, such practices were not compatible with our study. Therefore, the outcomes of our study were determined at the time of hospital discharge. Fourth, the tIVH group demonstrated a greater clinical severity than the non-tIVH group in the original cohort. Since the disease severity was still slightly higher after IPTW adjustment, the tIVH group may be worse in the unmeasured covariates. Fifth, we do not routinely performed MRI in tIVH group; thus, we are unable to evaluate the correlations between tIVH and DAI. Sixth, although we routinely monitored ICP values and Marshall scores for TBI patients requiring neurocritical care, our database did not record these data. Lastly, external validation of the Traumatic Graeb score in an independent patient group is necessary before it can be used to precisely predict outcomes.

## 5. Conclusions

In our cohort, the presence of IVH in TBI was associated with increased mortality and disability at discharge, regardless of associated injury. The Traumatic Graeb Score could be a reliable scale with favorable prognostic accuracy for the evaluation of the severity of IVH. A Traumatic Graeb Score ≥ 4 is an optimal cutoff value for poor short-term functional outcomes.

## Figures and Tables

**Figure 1 jcm-11-02127-f001:**
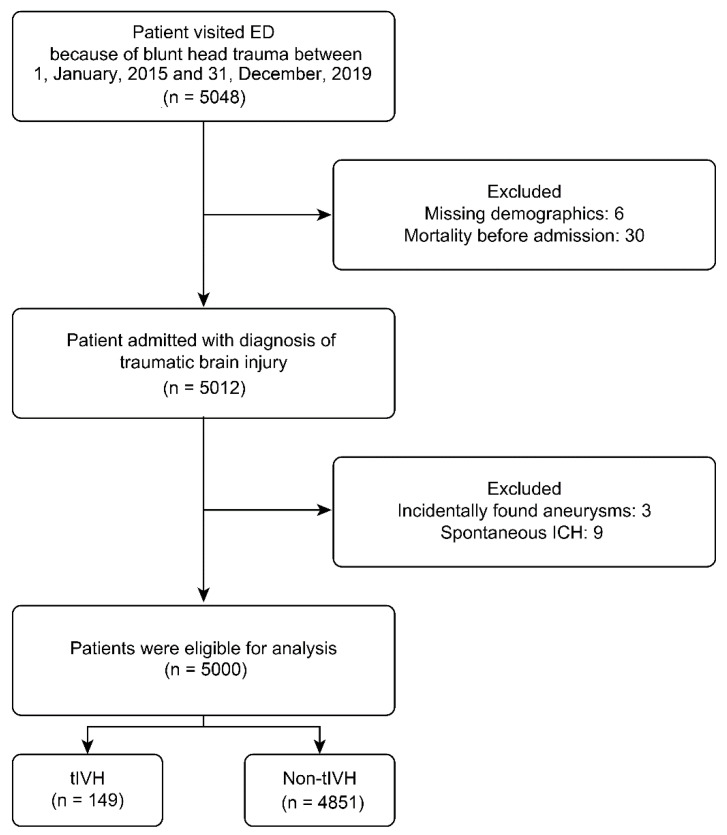
Flowchart of the current study and inclusion criteria. ED: emergency department; ICH: intracerebral hemorrhage; tIVH: traumatic intraventricular hemorrhage.

**Figure 2 jcm-11-02127-f002:**
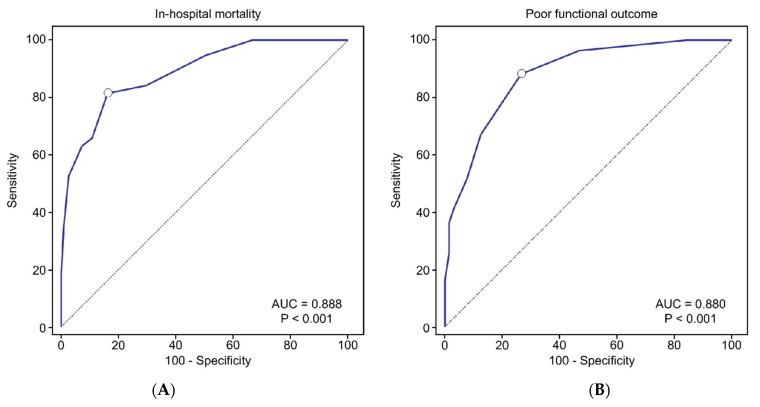
Receiver operating characteristic curves of the Traumatic Graeb Score in predicting in-hospital mortality (**A**) and poor functional outcome (**B**). AUC: area under the curve; mRS: modified Rankin Scale.

**Figure 3 jcm-11-02127-f003:**
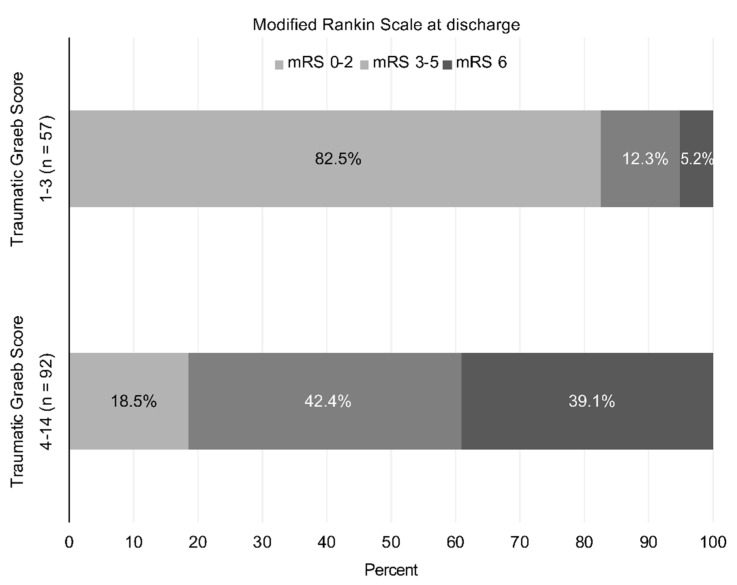
Outcomes at hospital discharge using modified Rankin Scale after traumatic intraventricular hemorrhage (tIVH).

**Table 1 jcm-11-02127-t001:** Demographics and characteristics of the study participants according to the presence or absence of tIVH (*n* = 5000).

	Before IPTW ^‡^				After IPTW ^†^			
Variable	Total (*n* = 5000)	tIVH (*n* = 149)	Non-tIVH (*n* = 4851)	STD	Total	tIVH	Non-tIVH	STD
Women	1706 (34.1)	41 (27.5)	1665 (34.3)	0.15	36.5	39.2	34.1	0.11
Age, years	49.5 ± 24.5	54.2 ± 26.0	49.4 ± 24.4	0.19	49.0 ± 26.4	48.4 ± 28.6	49.5 ± 24.3	−0.04
Comorbidity								
Stroke	343 (6.9)	10 (6.7)	333 (6.9)	−0.01	8.0	9.3	6.9	0.09
Hypertension	1485 (29.7)	50 (33.6)	1435 (29.6)	0.09	34.0	38.7	29.7	0.19
Coronary artery disease	484 (9.7)	20 (13.4)	464 (9.6)	0.12	12.0	14.6	9.7	0.15
Liver cirrhosis	83 (1.7)	3 (2.0)	80 (1.6)	0.03	1.7	1.6	1.7	<0.01
Diabetes mellitus	810 (16.2)	20 (13.4)	790 (16.3)	−0.08	18.3	20.6	16.2	0.11
Cancer	198 (4.0)	7 (4.7)	191 (3.9)	0.04	4.00	4.05	3.96	<0.01
GCS at ED	12.4 ± 4.0	9.1 ± 4.4	12.5 ± 3.9	−0.83	12.3 ± 4.0	12.1 ± 4.0	12.4 ± 4.0	−0.09
Systolic blood pressure at ED, mmHg	140.4 ± 32.2	145.4 ± 39.0	140.2 ± 32.0	0.14	140.4 ± 33.8	140.5 ± 35.5	140.4 ± 32.2	<0.01
Shock at ED	176 (3.5)	10 (6.7)	166 (3.4)	0.15	3.2	2.8	3.5	−0.04
Trauma scale								
ISS	19.7 ± 9.3	26.1 ± 9.6	19.5 ± 9.3	0.70	20.4 ± 8.9	21.2 ± 8.2	19.7 ± 9.4	0.18
NISS	22.4 ± 10.5	28.9 ± 10.8	22.2 ± 10.4	0.63	23.0 ± 10.2	23.7 ± 9.9	22.4 ± 10.5	0.13
Other Brain insults								
Skull Fracture	807 (16.1)	23 (15.4)	784 (16.2)	−0.02	18.0	20.1	16.1	0.10
Epidural hematoma	838 (16.8)	18 (12.1)	820 (16.9)	−0.14	14.5	12.0	16.7	−0.14
Subdural hematoma	2298 (46.0)	77 (51.7)	2221 (45.8)	0.12	48.5	51.2	46.0	0.10
Subarachnoid hemorrhage	2163 (43.3)	89 (59.7)	2074 (42.8)	0.34	46.2	49.4	43.3	0.12
Intracerebral hemorrhage	1270 (25.4)	75 (50.3)	1195 (24.6)	0.56	26.3	27.3	25.4	0.04

Abbreviations: tIVH, traumatic intraventricular hemorrhage; IPTW, inverse probability treatment weighting; STD, standardized difference; GCS, Glasgow Coma Scale; ED, emergency department; ISS, injury severity score; NISS, new injury severity scale; ^‡^ Data are presented as frequencies (percentages) or means ± standard deviations; ^†^ Data are presented as percentages or means ± standard deviations.

**Table 2 jcm-11-02127-t002:** Association between the presence of tIVH and outcomes (*n* = 5000).

	Before IPTW ^‡^		After IPTW ^†^			
Outcome	tIVH (*n* = 149)	Non-tIVH (*n* = 4851)	tIVH	Non-tIVH	OR or *B* (95% CI)	*p*
In-hospital mortality	38 (25.5)	427 (8.8)	11.4	9.2	1.27 (1.11–1.45)	<0.001
mRS > 2	85 (57.0)	491 (10.1)	37.9	10.6	5.13 (4.60–5.71)	<0.001
Operations						
External Ventricular Drainage	46 (30.9)	343 (7.1)	26.2	7.3	4.48 (3.95–5.08)	<0.001
Craniotomy	28 (18.8)	759 (15.6)	12.1	15.9	0.73 (0.65–0.82)	<0.001
Craniectomy	5 (3.4)	69 (1.4)	1.7	1.5	1.16 (0.84–1.60)	0.371
In-hospital complications						
New onset seizure	16 (10.7)	176 (3.6)	6.7	3.8	1.85 (1.53–2.23)	<0.001
New onset arrythmia	3 (2.0)	21 (0.4)	2.2	0.4	4.94 (3.12–7.83)	<0.001
Pneumonia	100 (67.1)	1223 (25.2)	42.8	26.2	2.10 (1.93–2.29)	<0.001
Gastrointestinal hemorrhage	5 (3.4)	51 (1.1)	1.0	1.1	0.92 (0.62–1.36)	0.667
Urinary tract infection	25 (16.8)	242 (5.0)	10.0	5.1	2.05 (1.75–2.41)	<0.001
Sepsis	12 (8.1)	110 (2.3)	8.0	2.4	3.57 (2.88–4.41)	<0.001
Admission days	20.9 ± 18.0	11.8 ± 46.2	18.6 ± 17.4	12.0 ± 45.9	6.64 (5.22, 8.07)	<0.001
ICU days	9.4 ± 11.0	3.2 ± 6.1	8.1 ± 10.4	3.3 ± 6.2	4.77 (4.43, 5.11)	<0.001

Abbreviations: tIVH, traumatic intraventricular hemorrhage; IPTW, inverse probability treatment weighting; OR, odds ratio; *B*, regression coefficient; CI: confidence interval; mRS, modified Rankin Scale; ICU, intensive care unit; ^‡^ Data are presented as frequency (percentage) or mean ± standard deviation; ^†^ Data are presented as percentage or mean ± standard deviation.

**Table 3 jcm-11-02127-t003:** Univariate analysis for predictors of in-hospital mortality and poor outcomes at discharge in patients with tIVH.

	In-Hospital Mortality			POOR Outcomes at Discharge		
Variables	Alive (*n* = 111)	Dead (*n* = 38)	*p*	mRS ≤ 2 (*n* = 64)	mRS > 2 (*n* = 85)	*p*
Women (*n*, %)	32 (28.8%)	9 (23.7%)	0.675 ^†^	23 (35.9%)	18 (21.2%)	0.063 ^†^
Age ≥ 65 years old (*n*, %)	43 (38.7%)	20 (52.6%)	0.183 ^†^	19 (29.7%)	44 (51.8%)	0.008 ^†^
Stroke (*n*, %)	8 (7.2%)	2 (5.3%)	1.000 ^†^	3 (4.7%)	7 (8.2%)	0.516 ^†^
Hypertension (*n*, %)	37 (33.3%)	13 (34.2%)	1.000 ^†^	16 (25%)	34 (40%)	0.079 ^†^
Coronary artery disease (*n*, %)	13 (11.7%)	7 (18.4%)	0.285 ^†^	5 (7.8%)	15 (17.6%)	0.094 ^†^
Liver cirrhosis (*n*, %)	2 (18%)	1 (2.6%)	1.000 ^†^	2 (3.1%)	1 (1.2%)	0.577 ^†^
Diabetes mellitus (*n*, %)	15 (13.5%)	5 (13.2%)	1.000 ^†^	6 (9.4%)	14 (16.5%)	0.235^†^
ED GCS severity score ^‡^	0.7 ± 0.7	1.3 ± 0.6	<0.001 *	0.5 ± 0.5	1.2 ± 0.7	<0.001 *
Shock at ED (*n*, %) ^**^	6 (5.4%)	4 (10.5%)	0.277 ^†^	2 (3.1%)	8 (9.4%)	0.189 ^†^
Skull fracture (*n*, %)	17 (15.3%)	6 (15.8%)	1.000 ^†^	9 (14.1%)	14 (16.5%)	0.820 ^†^
Epidural hematoma (*n*, %)	13 (11.7%)	5 (13.2%)	0.779 ^†^	10 (15.6%)	8 (9.4%)	0.312 ^†^
Subdural hematoma (*n*, %)	56 (50.5%)	21 (55.3%)	0.708 ^†^	29 (45.3%)	48 (56.5%)	0.189 ^†^
Subarachnoid hemorrhage (*n*, %)	66 (59.5%)	23 (60.5%)	1.000 ^†^	39 (60.9%)	50 (58.8%)	0.867 ^†^
Intracerebral hemorrhage (*n*, %)	57 (51.4%)	21 (55.23)	0.710 ^†^	27 (42.2%)	51 (60%)	0.033 ^†^
The Graeb score	2.6 ± 1.6	6.3 ± 2.9	<0.001 *	2.1 ± 1.3	4.7 ± 2.7	<0.001 *

Numerical data: mean ± standard deviation Nominal data: *n* (%). * Student’s *t*-test test ^†^ Chi-square test; Abbreviations: tIVH, traumatic intraventricular hemorrhage; mRS, modified Rankin Scale; ED, emergency department; GCS, Glasgow Coma Scale. ^‡^ ED GCS severity score: according to severity, ED GCS was divided into three groups: grade 0 (GCS 13–15), grade 1 (GCS 5–12), and grade 2 (GCS 3–4). ** Shock at ED: Systolic blood pressure at ED < 90mmHg.

**Table 4 jcm-11-02127-t004:** Multivariate analysis for predictors of in-hospital mortality and poor outcomes at discharge in patients with tIVH.

	In-Hospital Mortality		Poor Outcomes at Discharge (mRS > 2)	
Variable	OR (95% CI)	*p **	OR (95% CI)	*p **
Age ≥ 65	2.95 (1.02–8.56)	0.046	4.64 (1.56–13.80)	0.006
ED GCS severity score ^†^	2.95 (1.41–6.17)	0.004	6.65 (2.62–16.89)	0.001
The Graeb score	1.84 (1.48–2.30)	<0.001	1.99 (1.47–2.71)	<0.001
Sex	N/A	N/A	-	0.594
Hypertension	N/A	N/A	-	0.873
Coronary artery disease	N/A	N/A	-	0.120
Shock at ED ^‡^	N/A	N/A	-	0.498
Subdural hematoma	N/A	N/A	-	0.123
Intracerebral hemorrhage	N/A	N/A	-	0.244

* Multivariate logistic regression; Abbreviations: tIVH, traumatic intraventricular hemorrhage; mRS, modified Rankin Scale; OR, odds ratio; CI: confidence interval; ED, emergency department; GCS, Glasgow Coma Scale. ^†^ ED GCS severity score: according to severity, ED GCS was divided into three groups: grade 0 (GCS 13–15), grade 1 (GCS 5–12), and grade 2 (GCS 3–4). ^‡^ Shock at ED: Systolic blood pressure at ED < 90 mmHg.

**Table 5 jcm-11-02127-t005:** Components of the novel Traumatic Graeb Score.

Components	Traumatic Graeb Score Points
GCS score	
13–15	0
5–12	1
3–4	2
Age	
<65 years old	0
≥65 years old	1
IVH grading	
Lateral Ventricles (Right and Left, calculated seperately)	
Trace of blood or mild bleeding	1
Less than 50% of ventricle with blood	2
More than 50% of ventricle with blood	3
Filled with blood and expanded	4
3rd ventricle	
No blood	0
Blood presents, ventricle size normal	1
Filled with blood and expanded	2
4th ventricle	
No blood	0
Blood presents, ventricle size normal	1
Filled with blood and expanded	2
Total Traumatic Graeb Score: 1–15	

Abbreviations: GCS, Glasgow Coma Scale; IVH, intraventricular hemorrhage.

## Data Availability

Not applicable.

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
