# Peer review of "The Role of Intraventricular Hemorrhage in Traumatic Brain Injury: A Novel Scoring System"

_jcm, 2022, doi:10.3390/jcm11082127_

Round 1

Reviewer 1 Report

The authors developed a new score for traumatic IVH, a novel Graeb Score, which is associated with higher mortality and poor functional outcome. The study is interesting and the score is simple and useful for clinical assessment.

The methods and the statistical assessment are clear.

There are just a few minor points:

1.) Two experienced neuroradiologists evaluated the CT scans.

Can you provide more information about the expertise ( years working in a neuroradiology department)?

Did you perform statistical evaluation of intra- or interobserver agreement? If not, why ??

2.) It is described, that tIVH patients had lower rates of craniotomy, is it possible to provide an explanation??

3.) Why was th typical DAI pattern not evaluated in CT scans? I thought there is an association between DAI and intraventricular hemorrhage on initial CT scans ?

Reviewer 2 Report

The authors retrospectively reviewed the records of 5048 victims of traumatic brain injury (TBI). The aim of the study was (1) to assess the association of traumatic intraventricular hemorrhage (tIVH) with in-hospital mortality and poor functional outcome following TBI and (2) to propose the validation of a novel scoring system (Traumatic Graeb Score) to predict poor short-term functional outcomes associated to tIVH.

This interesting subject may improve outcome prediction and the clinical decision-taking process following TBI, particularly in tIVH patients. However, the study design and manuscript present points that raise concerns and require clarification.  

Comments:

#1 – The abstract does not assert clearly the objectives of the study, does not report the study design nor the sample size of tIVH patients group. Furthermore, the results presented in the abstract are confusing. Therefore, this section of the manuscript needs to be improved.

# 2 – Regarding inclusion and exclusion criteria, the authors should present the study design (case-control ?) as well as the sample size calculation.

#3 – Considering the exclusion criteria (loss of patients) reported due to missing demographics, mortality before admission (possibly the most severe cases), incidentally found aneurysms and spontaneous ICH (possibly with IVH), the authors should consider and discuss this as a possible study bias and limitation. Although loss of 48 patients seems insignificant if we ponder a sample of 5048 patients, if we analyze specifically the tIVH sample size (n=149), the reported loss of patients (nearly one third of possible tIVH cases) is significant and may have affected the Traumatic Groeb scoring system validation.

#4 – tIVH rate is higher in more severe patients, as corroborated in this study by the higher mean GCS scores in tIVH group of patients (Table 1). The authors should detail the frequency of severe TBI patients in each group - with tIVH and non-tIVH, allowing the reader to reflect about the heterogeneity of TBI severity between the groups (they presented only the means of GCS scores). Indeed, reinforcing the greater clinical severity of the tIVH group, the frequency of shock, mass lesions as well as ISS scores (an indicative of multisystem trauma) were higher in this group. In addition, the tIVH group presented higher hospital length of stay and rate of clinical complications. Therefore, the clear difference in clinical severity between the groups (tIVH or non-tIVH) may possibly have affected the reliability of the statistical adjustment analysis applied.

#5 – The authors should explain why they did not investigate potential associations with outcomes in all the TBI patients studied, n=5012 (tIVH and non-tIVH), preceding the analysis of the tIVH group exclusively.

#6 – Taking into account that rate of neurosurgical procedures presented association with the outcomes studied; the authors should state why they did not include ICP values, nor the Marshall scores, as variables in the study. These are important routine parameters monitored in neurocritical care.

#7 – In the discussion (pg12, 2 paragraph), the authors mention that …“We proposed that tIVH presented initially would be a key contributor to patient’s outcomes at discharge regardless of associated injuries”…This statement does not seem to be supported by the results presented.

#8 – Finally, the authors mention that informed consent statement “was obtained from all” the patients. They should detail how this was attained, since the study was retrospective and there was an important in-hospital mortality.

Reviewer 3 Report

The authors divided patients with TBI into two groups: one arm with tIVH and the another arm without IVH. They reported age >65y, GCS and Graeb score as independent predictors for mortality and poor outcome at discharge. By using those predictors, a traumatic Graeb score system was developed. For the balance of both groups, an inverse probability of treatment was used. All in all, the key message of this study is clear with adequate methodical analysis. Here are some concerns:

1. The authors described a traumatic graeb score system with a point range of 1-14. However, the outcome interpretation is not quite clear to me. Patients with a point range of 1-3 have over 80% favorable outcome whereas patients with a point range of 4-14 have less than 20% favorable outcome. I think that the range between 4-14 is too large to put them together. I would suggest to make a graphic or table showing outcome or mortality divided by each points.

2. As the author mentioned shortly in the limitation part, the study only described the outcome at hospital discharge. This inherits a major limitation concerning a score system. We all know that patients with tIVH need more frequently drains and longer ICU stay with associated complications. Thus, those patients need more time to recover from the TBI. A score system is interesting to evaluate the possibility of recovery of those patients. If the authors could add the outcome data at 3- or 6months follow-up, the quality of this study will be more increased.

Round 2

Reviewer 2 Report

The authors attended most of the comments while presented the points that were criticized as limitations of the study.